**Original Manuscript ID:** 1215

Original Article Title: "A spatiotemporal graph neural network with multi granularity for air quality prediction"

**To:** ICLR2023

**Re:** Response to reviewers

Dear Editors and Reviewers,

We would like to thank you for the time devoted to handling and reviewing our manuscript "A spatiotemporal graph neural network with multi granularity for air quality prediction" submitted to ICLR 2023.

We are particularly grateful for the constructive comments raised by the reviewers for the manuscript, with an opportunity to address the reviewers' comments. We believe that the quality and readability of the revised manuscript have been significantly improved thanks to these valuable comments.

We are uploading (a) our point-by-point response to the comments (below) (response to reviewers) and (b) an updated manuscript.

# Reviewer 1

**Reviewer tjqh,** Strength And Weaknesses:

Strength:
1. The paper addresses an important problem in air pollutant concentration prediction. By integrating the mechanical model and machine learning, the work bridges the traditional methods and current trend.
2. The high-level presentation is good which is easy to follow and understand.
3. The experiment is solid and comprehensive.

   **Author response:** We are very grateful for your comments and recognition of the strengths of the manuscript.

Weakness:
  1. The paper uses inconsistent terms and symbols, which is confusing at some time For example, R and r_a are to represent the region in section 3.1, but authors use the same symbol for messages in equation 2, which is a little ambiguous. In figure2, the input data doesn't correspond to the variables in the definition section, I would suggest the authors to use consistent symbolic notations for the important features in the both the text and the explanatory figures. Some of the figures need more explanation, e.g., in figure1, the meanings of the axes are unknown. Typo: neighbor node net->neighbor node set

2 The novelty is not outstanding. Although integrating the mechanical model and machine learning is interesting, most of the techniques are well-established and the authors simply apply them to the new problem. The overall framework is the same as this paper: Jiahui Xu, Ling Chen, Mingqi Lv, Chaoqun Zhan, Sanjian Chen, and Jian Chang. HighAir: A hierarchical graph neural network-based air quality forecasting method. arXiv preprint arXiv: 2101.04264, 2021.

   **Author response:**

1. We thank the reviewer for this comment. We have revised the issues of inconsistent terms and symbols, figures explanation, typo and so on. We believe that the quality and readability of the revised manuscript have been significantly improved thanks to these valuable comments.

2. The main contributions of this paper include:

   • We innovatively propose a dynamic spatiotemporal graph model combining mechanism model and graph neural network. The adjacency matrix and edge weight vector of dynamic graph are constructed based on the simulation results of diffusion, transport and deposition of polluted air mass by mechanism model, so that the

architecture learns the spatial influence relationship among multi granularity stations.

- In order to better learn the temporal patterns and periodicity of pollutants, we propose to add time characteristic attributes of quarter, month, week, hour and holiday to each node in the encoder, use LSTM based on attention mechanism for temporal learning in the decoder to enhance MGST-GNN and use a new data augmented method to solve the problem of insufficient training data.

As stated in the paper, our model framework refers to HighAir, but our method is very different from Highair:

1. A static graph is built in HighAir, while a dynamic graph is built by HYSPLIT. Our dynamic graphs are more conducive to learning spatial relationships.

2. In terms of the attributes of graph nodes, we have different geographical features and add time features.

3. For sequential relationship capture, we use the LSTM of the attention mechanism.

Reviewer tjqh, **Clarity, Quality, Novelty And Reproducibility:**
The techniques used in this paper are solid. The overall presentation is good and clear. The originality is average.

**Author response:** We greatly appreciate your positive comments and recognition of the paper.

Reviewer tjqh, **Summary Of The Review:**
This paper is well-written and addresses an important question possibly helping bridge environmental science and machine learning. The work falls short on the technical novelty. However, it's still marginally above the acceptance line.

**Author response:** We greatly appreciate your positive comments and recognition of the paper.

Reviewer tjqh, **Correctness:**
3: Some of the paper's claims have minor issues. A few statements are not well-supported, or require small changes to be made correct.

**Author response:** We are very grateful for your comments on the manuscript. We have made a lot of endeavors to enhance the quality of the manuscript, and revised the manuscript for many times word by word and sorry for that there have

minor issues that make the paper hard to read. We carried out a careful revision which can be seen in the revised manuscript and we hope it reach your standard.

Reviewer tjqh, **Technical/ Empirical Novelty And Significance:**
2: The contributions are only marginally significant or novel.

**Author response:** The main contributions of this paper include:

• We innovatively propose a dynamic spatiotemporal graph model combining mechanism model and graph neural network. The adjacency matrix and edge weight vector of dynamic graph are constructed based on the simulation results of diffusion, transport and deposition of polluted air mass by mechanism model, so that the architecture learns the spatial influence relationship among multi granularity stations.

• In order to better learn the temporal patterns and periodicity of pollutants, we propose to add time characteristic attributes of quarter, month, week, hour and holiday to each node in the encoder, use LSTM based on attention mechanism for temporal learning in the decoder to enhance MGST-GNN and use a new data augmented method to solve the problem of insufficient training data.

To our knowledge, no one has yet used the professional air quality model HYSPLIT to dynamically construct the graph structure. Many STGNNs [1, 2, 3] depend on a pre-defined graph to indicate the relationship between nodes. However, such a graph is not available or is incomplete in many cases. An intuitive idea is to train an adjacency matrix indicates the dependency among nodes. However, since the learning of graph structure and STGNNs are coupled compactly, and there is no supervised loss information for graph structure learning [4], optimizing such a contiguous matrix usually leads to a complex bilevel optimization problem [5]. The most important thing is that the adjacency matrix is learned from historical data and cannot adapt smoothly and dynamically according to the change of field conditions. Fortunately, we can alleviate these problems based on the HYSPLIT. We aim to learn a dynamic graph from field impact factor such as meteorological and topographic conditions, which can be easily extended to other spatiotemporal forecasting tasks. For example, in water quality prediction, a professional hydrodynamic model (MIKE) can be used to dynamically construct the graph structure, so as to better learn the

influence of water quality in different regions on the prediction points. What is more, in the prediction of traffic flow, professional Traffic Flow Dynamics Model (Traffic Wave Models) can be used to dynamically build the graph structure, so as to better learn the influence of different traffic intersections on the predicted points.

[1] Yaguang Li, Rose Yu, Cyrus Shahabi, and Yan Liu. 2018. Diffusion Convolutional Recurrent Neural Network: Data-Driven Traffic Forecasting. In ICLR.

[2] Zonghan Wu, Shirui Pan, Guodong Long, Jing Jiang, and Chengqi Zhang. 2019. Graph WaveNet for Deep Spatial-Temporal Graph Modeling. In IJCAI.

[3] Chuanpan Zheng, Xiaoliang Fan, Cheng Wang, and Jianzhong Qi. 2020. GMAN: A Graph Multi-Attention Network for Traffic Prediction. In AAAI.

[4] Haozhe Lin, Yushun Fan, Jia Zhang, and Bing Bai. 2021. REST: Reciprocal Framework for Spatiotemporal-coupled Predictions. In The Web Conference.

[5] Luca Franceschi, Mathias Niepert, Massimiliano Pontil, and Xiao He. 2019. Learning Discrete Structures for Graph Neural Networks. In ICML.

In summary, the main innovations of this paper are the construction of dynamic graph neural network of spatiotemporal data and the innovative application of air quality prediction combining machine learning model and mechanism model. The theme and innovation of this paper are in line with the conference topic and subject areas: applications in audio, speech, robotics, neuroscience, biology, or any other field.

# Reviewer 2

Reviewer VRie,  **Strength And Weaknesses:**

Strength:
1. The proposed model is easy to understand and the idea behind partial graph and global graph is explained well.
2. The experimental results contain many ablation study.

**Author response:** We are very grateful for your comments and recognition of the strengths of the manuscript.

Weaknesses:
1. The contents of this paper definitely can be organized better.
- The experimental results are all in appendixes. The experiments section in main body only have some vague description of the model's performance, e.g. "the MAE ... decreases about 1.88 and 1.52 ...", or "MAE decreases about 1.99, 2.09, 0.59...". This is confusing and it's hard for a general reader to understand what these metrics mean unless they go to check the tables in the appendix. I'd suggest to make the paper's main body self-contained.
- The authors listed many related models in the related work section. However, the differences between existing models and the proposed model were not discussed thoroughly.
2. The experimental results were not presented well. For table 6 in appendix A.6, the first two rows (method MGST_GNN) are all bolded. However, it doesn't seem that MGST_GNN gets the best metric in each column. For example, under metric=1h, HighAir's MAE is 7.12 while MGST_GNN's is 7.16. Under metric=18h, MGST_GNN's MAE is 27.92 while HighAir's is 27.49 and ST-UNet's is 27.86. I think the tradition is to bold the best metric in the corresponding column.
3. In Appendix A.6 the authors claimed that MGST_GNN outperforms the baseline models and provided their explanations. I don't think this is very convincing. Even if MGST_GNN gets better results in Table 6, it doesn't imply that, for example, MGST_GNN outperforms HighAir model because MGST_GNN uses HYSPLIT model. A more promising way is to provide ablation study results on the same dataset.
4. The proposed model uses HYSPLIT to construct the graphs. Since HYSPLIT is specifically designed for air quality related tasks, how does MGST_GNN generalize to other spatiotemporal forecasting tasks? It would be good to see some discussions along this line.

**Author response:**

1. We have reorganized the paper, and we believe that the quality and readability of the revised manuscript have been significantly improved thanks to these valuable comments. Specifically, the experimental part of the appendix was moved to the paper's main body. At the same time, in order to control the paper's length, the original sections 1 and 2 were merged, and the data enhancement experiment remains in the appendix.

2. Thank you very much for pointing out the error of bold display of the experimental results. We have carefully revised the relevant problems.

3. In order to further prove the validity of our proposed MGST_GNN model (fusion mechanism model HYSPLIT and machine learning STGNN), an ablation experiment analysis is provided in the appendix.

4. In this paper, we are the first to use the professional model HYSPLIT to build the graph dynamically, and we have proved its effectiveness through experiments. This method eliminates the construction process of specially designed auxiliary network to learn the edges of graph and provides a new way for the construction of graph neural network, which can be easily extended to other spatiotemporal forecasting tasks. For example, in water quality prediction, a professional hydrodynamic model (MIKE) can be used to dynamically construct the graph structure, so as to better learn the influence of water quality in different regions on the prediction points. What is more, in the prediction of traffic flow, professional Traffic Flow Dynamics Model (Traffic Wave Models) can be used to dynamically build the graph structure, so as to better learn the influence of different traffic intersections on the predicted points.

Reviewer VRie, **Clarity, Quality, Novelty And Reproducibility:**
I think in its current form, this paper has quite a lot of improvement space for its clarity and quality. It seems that both codes and datasets are not open sourced, I'm not confident about its reproducibility.

Author response: We have made a lot of endeavors to enhance the quality of the manuscript，and we believe that the quality and readability of the revised manuscript have been significantly improved thanks to these valuable comments. The code is released on GitHub, which it provides a way to obtain datasets. Because of anonymous review, it was not shown in the manuscript.

Reviewer VRie, **Summary Of The Review:**
There are a lot of improvements that can be made in this paper's organizations, experimental results presentations and discussions. I think it's not ready yet.

**Author response:** We have reorganized the paper and made a lot of endeavors to enhance the quality of the manuscript. we believe that the quality and readability of the revised manuscript have been significantly improved thanks to these valuable comments.

Reviewer VRie, **Correctness:**

3: Some of the paper's claims have minor issues. A few statements are not well-supported, or require small changes to be made correct.

**Author response:** We are very grateful for your comments on the manuscript. We have made a lot of endeavors to enhance the quality of the manuscript, and revised the manuscript for many times word by word and sorry for that there have minor issues that make the paper hard to read. We carried out a careful revision which can be seen in the revised manuscript and we hope it reach your standard.

Reviewer VRie, **Technical/ Empirical Novelty And Significance:**

2: The contributions are only marginally significant or novel.

**Author response:** The main contributions of this paper include:

• We innovatively propose a dynamic spatiotemporal graph model combining mechanism model and graph neural network. The adjacency matrix and edge weight vector of dynamic graph are constructed based on the simulation results of diffusion, transport and deposition of polluted air mass by mechanism model, so that the architecture learns the spatial influence relationship among multi granularity stations.

• In order to better learn the temporal patterns and periodicity of pollutants, we propose to add time characteristic attributes of quarter, month, week, hour and holiday to each node in the encoder, use LSTM based on attention mechanism for temporal learning in the decoder to enhance MGST-GNN and use a new data augmented method to solve the problem of insufficient training data.

To our knowledge, no one has yet used the professional air quality model HYSPLIT to dynamically construct the graph structure. Many STGNNs [1, 2, 3] depend on a pre-defined graph to indicate the relationship between nodes. However, such a graph is not available or is incomplete in many cases. An intuitive idea is to train an adjacency matrix indicates the dependency among nodes. However, since the learning of graph structure and STGNNs are coupled compactly, and there is no supervised loss information for graph structure learning [4], optimizing such a contiguous matrix usually leads to a complex bilevel optimization problem [5]. The most important thing is that the adjacency matrix is learned from historical data and cannot adapt smoothly and dynamically according to the change of field conditions.

Fortunately, we can alleviate these problems based on the HYSPLIT. We aim to learn a dynamic graph from field impact factor such as meteorological and topographic conditions, which can be easily extended to other spatiotemporal forecasting tasks. For example, in water quality prediction, a professional hydrodynamic model (MIKE) can be used to dynamically construct the graph structure, so as to better learn the influence of water quality in different regions on the prediction points. What is more, in the prediction of traffic flow, professional Traffic Flow Dynamics Model (Traffic Wave Models) can be used to dynamically build the graph structure, so as to better learn the influence of different traffic intersections on the predicted points.

[1] Yaguang Li, Rose Yu, Cyrus Shahabi, and Yan Liu. 2018. Diffusion Convolutional Recurrent Neural Network: Data-Driven Traffic Forecasting. In ICLR.

[2] Zonghan Wu, Shirui Pan, Guodong Long, Jing Jiang, and Chengqi Zhang. 2019. Graph WaveNet for Deep Spatial-Temporal Graph Modeling. In IJCAI.

[3] Chuanpan Zheng, Xiaoliang Fan, Cheng Wang, and Jianzhong Qi. 2020. GMAN: A Graph Multi-Attention Network for Traffic Prediction. In AAAI.

[4] Haozhe Lin, Yushun Fan, Jia Zhang, and Bing Bai. 2021. REST: Reciprocal Framework for Spatiotemporal-coupled Predictions. In The Web Conference.

[5] Luca Franceschi, Mathias Niepert, Massimiliano Pontil, and Xiao He. 2019. Learning Discrete Structures for Graph Neural Networks. In ICML.

In summary, the main innovations of this paper are the construction of dynamic graph neural network of spatiotemporal data and the innovative application of air quality prediction combining machine learning model and mechanism model. The theme and innovation of this paper are in line with the conference topic and subject areas: applications in audio, speech, robotics, neuroscience, biology, or any other field.

**Reviewer 3**

**Reviewer mQij,** Strength And Weaknesses:

Strength:
The topic of air quality prediction is of great social impact.

**Author response:** We are very grateful for your comments and recognition of the strengths of the manuscript.

Weaknesses:
1. The format of the paper is unprofessional. All the experimental results are in the appendix.
2. Figures are in poor resolution. Please use vectorized images for professional academic writing.
3.The technical contribution is limited. The settings of LSTM, GNN, and attention mechanism are commonly used for air quality prediction. The motivation of multi-granularity also has been studied thoroughly for spatial-temporal forecasting.
4. The experiment cannot fully support the validity of the model. Air quality prediction is inherently a time series prediction task. Time series forecasting models, such as N-BEATS, autoformer, informer, etc, are expected to compare.

**Author response:**

1. We have reorganized the paper, and we believe that the quality and readability of the revised manuscript have been significantly improved thanks to these valuable comments. Specifically, the experimental part of the appendix was moved to the paper's main body. At the same time, in order to control the paper's length, the original sections 1 and 2 were merged, and the data enhancement experiment remains in the appendix.

2. Thank you very much for your reminding. We have provided higher resolution illustrations in the revised manuscript. Once the paper is accepted, we will also provide the original drawing.

3. In this paper, we integrate the mechanism model and the popular machine learning model organically for the first time, and overcome the shortcomings of the mechanism model and machine learning respectively. The main contributions of this paper include:

• We innovatively propose a dynamic spatiotemporal graph model combining mechanism model and graph neural network. The adjacency matrix and edge weight vector of dynamic graph are constructed based on the simulation results of diffusion, transport and deposition of polluted air mass by mechanism model, so that the

architecture learns the spatial influence relationship among multi granularity stations.

- In order to better learn the temporal patterns and periodicity of pollutants, we propose to add time characteristic attributes of quarter, month, week, hour and holiday to each node in the encoder, use LSTM based on attention mechanism for temporal learning in the decoder to enhance MGST-GNN and use a new data augmented method to solve the problem of insufficient training data.

To our knowledge, no one has yet used the professional air quality model HYSPLIT to dynamically construct the graph structure. Many STGNNs depend on a pre-defined graph to indicate the relationship between nodes. However, such a graph is not available or is incomplete in many cases. An intuitive idea is to train an adjacency matrix indicates the dependency among nodes. However, since the learning of graph structure and STGNNs are coupled compactly, and there is no supervised loss information for graph structure learning, optimizing such a contiguous matrix usually leads to a complex bilevel optimization problem. The most important thing is that the adjacency matrix is learned from historical data and cannot adapt smoothly and dynamically according to the change of field conditions. Fortunately, we can alleviate these problems based on the HYSPLIT. We aim to learn a dynamic graph from field impact factor such as meteorological and topographic conditions, which can be easily extended to other spatiotemporal forecasting tasks. For example, in water quality prediction, a professional hydrodynamic model (MIKE) can be used to dynamically construct the graph structure, so as to better learn the influence of water quality in different regions on the prediction points. What is more, in the prediction of traffic flow, professional Traffic Flow Dynamics Model (Traffic Wave Models) can be used to dynamically build the graph structure, so as to better learn the influence of different traffic intersections on the predicted points.

4. The method in this paper is based on spatiotemporal graph neural network, and its main innovation point is how to construct dynamic graph. Therefore, in the experimental part, due to the limitation of the length of the conference paper, we only compared with the current advanced graph model.

Reviewer mQij, **Clarity, Quality, Novelty And Reproducibility:**

The writing is easy to follow. But the novelty is limited. We have concerns about the reproducibility, because the hyperparameter setting is not provided.

Author response: We have made a lot of endeavors to enhance the quality of the manuscript，including innovative statement and experimental super-parameter setting. The code is released on GitHub. Because of anonymous review, it was not shown in the manuscript.

Reviewer mQij, **Summary Of The Review:**

The topic is important. But the novelty is limited, the experiment is weak. The paper needs lots of effort to improve.

**Author response:** We have reorganized the paper and made a lot of endeavors to enhance the quality of the manuscript. We believe that the quality and readability of the revised manuscript have been significantly improved thanks to these valuable comments.

Reviewer mQij, **Correctness:**

3: Some of the paper's claims have minor issues. A few statements are not well-supported, or require small changes to be made correct.

**Author response:** We are very grateful for your comments on the manuscript. We have made a lot of endeavors to enhance the quality of the manuscript, and revised the manuscript for many times word by word and sorry for that there have minor issues that make the paper hard to read. We carried out a careful revision which can be seen in the revised manuscript and we hope it reach your standard.

Reviewer mQij, **Technical/ Empirical Novelty And Significance:**

2: The contributions are only marginally significant or novel.

**Author response:** The main contributions of this paper include:

• We innovatively propose a dynamic spatiotemporal graph model combining mechanism model and graph neural network. The adjacency matrix and edge weight vector of dynamic graph are constructed based on the simulation results of diffusion, transport and deposition of polluted air mass by mechanism model, so that the architecture learns the spatial influence relationship among multi granularity stations.

• In order to better learn the temporal patterns and periodicity of pollutants, we propose to add time characteristic attributes of quarter, month, week, hour and

holiday to each node in the encoder, use LSTM based on attention mechanism for temporal learning in the decoder to enhance MGST-GNN and use a new data augmented method to solve the problem of insufficient training data.

To our knowledge, no one has yet used the professional air quality model HYSPLIT to dynamically construct the graph structure. Many STGNNs [1, 2, 3] depend on a pre-defined graph to indicate the relationship between nodes. However, such a graph is not available or is incomplete in many cases. An intuitive idea is to train an adjacency matrix indicates the dependency among nodes. However, since the learning of graph structure and STGNNs are coupled compactly, and there is no supervised loss information for graph structure learning [4], optimizing such a contiguous matrix usually leads to a complex bilevel optimization problem [5]. The most important thing is that the adjacency matrix is learned from historical data and cannot adapt smoothly and dynamically according to the change of field conditions. Fortunately, we can alleviate these problems based on the HYSPLIT. We aim to learn a dynamic graph from field impact factor such as meteorological and topographic conditions, which can be easily extended to other spatiotemporal forecasting tasks. For example, in water quality prediction, a professional hydrodynamic model (MIKE) can be used to dynamically construct the graph structure, so as to better learn the influence of water quality in different regions on the prediction points. What is more, in the prediction of traffic flow, professional Traffic Flow Dynamics Model (Traffic Wave Models) can be used to dynamically build the graph structure, so as to better learn the influence of different traffic intersections on the predicted points.

[1] Yaguang Li, Rose Yu, Cyrus Shahabi, and Yan Liu. 2018. Diffusion Convolutional Recurrent Neural Network: Data-Driven Traffic Forecasting. In ICLR.

[2] Zonghan Wu, Shirui Pan, Guodong Long, Jing Jiang, and Chengqi Zhang. 2019. Graph WaveNet for Deep Spatial-Temporal Graph Modeling. In IJCAI.

[3] Chuanpan Zheng, Xiaoliang Fan, Cheng Wang, and Jianzhong Qi. 2020. GMAN: A Graph Multi-Attention Network for Traffic Prediction. In AAAI.

[4] Haozhe Lin, Yushun Fan, Jia Zhang, and Bing Bai. 2021. REST: Reciprocal Framework for Spatiotemporal-coupled Predictions. In The Web Conference.

[5] Luca Franceschi, Mathias Niepert, Massimiliano Pontil, and Xiao He. 2019. Learning Discrete Structures for Graph Neural Networks. In ICML.

In summary, the main innovations of this paper are the construction of dynamic graph neural network of spatiotemporal data and the innovative application of air quality prediction combining machine learning model and mechanism model. The theme and innovation of this paper are in line with the conference topic and subject areas: applications in audio, speech, robotics, neuroscience, biology, or any other field.

We really appreciate the helpful comments of the referees and hope that we have produced a better account of our work now. We believe that the revised manuscript is acceptable for publication.

Sincerely yours,
The authors

# A SPATIOTEMPORAL GRAPH NEURAL NETWORK WITH MULTI GRANULARITY FOR AIR QUALITY PREDICTION

**Anonymous authors**

## ABSTRACT

Air quality prediction is a complex system engineering. How to fully consider the impact of meteorological, spatial and temporal factors on air quality is the core problem. To address this central conundrum, in an elaborate encoder-decoder architecture, we propose a new air quality prediction method based on multi-granularity spatiotemporal graph network. At the encoder, firstly, we use multi granularity graph and the well-known HYSPLIT model to build spatial relationship and dynamic edge relationship between nodes, respectively, while meteorological, temporal and topographic characteristics are used to build node features and LSTM (Long Short Term Memory) is used to learn the time-series relationship of pollutant concentration. At the decoder, secondly, we use the attention mechanism LSTM for decoding and forecasting of pollutant concentration. The proposed model is capable of tracking different influences on prediction resulting from the changes of air quality. On a project-based dataset, we validate the effectiveness of the proposed model and examine its abilities of capturing both fine-grained and long-term influences in pollutant process. We also compare the proposed model with the state-of-the-art air quality forecasting methods on the dataset of Yangtze River Delta city group, the experimental results show the appealing performance of our model over competitive baselines.

## 1 INTRODUCTION

Air quality which is closely related to human public health, has been a common research hotspot focused by scholars all over the world. At present, a large number of air quality monitoring stations (stations for short) have been built in major cities to monitor the concentration of air pollutants (PM2.5, PM10, $O_3$, etc.) and meteorological parameters (temperature, pressure, wind speed, wind direction, humidity, etc.). However, these stations only can monitor real-time air quality, and fail to provide air quality prediction (AQP) and auxiliary support for urban intelligent decision-making or activity planning. How to construct an AQP model using a large amount of historical monitoring data has become a hot research topic in the field of environmental engineering. Unfortunately, AQP is an extremely complex system engineering. On the one hand, air quality is related to pollutant emission, which is a type of time sequence and has periodicity; On the other hand, there exist physical and chemical changes of pollutants in the air, such as diffusion and deposition, which are greatly affected by meteorological and geographical locations; Finally, air quality also has certain probability, such as unexpected pollution leakage events will lead to a sharp decline in air quality.

The commonly used types of AQP include mechanism model (MM) and machine learning (ML) methods. The MM method Jittra et al. (2015); Zhang (2017); Arystanbekova (2004); Stein et al. (2015); Wang et al. (2012); Yi et al. (2018), also known as numerical model, uses atmospheric physical and chemical reactions to model the emission and diffusion process of air pollutants, and then carries out AQP. For example, Gaussian diffusion models of AERMOD and ADMS, Lagrange models of CALPUFF and HYSPLIT can be applied to small-scale and medium AQP Jittra et al. (2015); and the third generation air quality models Zhang (2017) such as CMAQ, CAMX, WRF-CHEM, NAQPMS, and so on can be applied to predict large-scale air quality. However, most of MM methods require many empirical parameters and assumptions, which are prone to be reliable for a specific environment but not for all urban environments Yi et al. (2018). For example, AERMOD is an empirical model which is mainly applicable to small-scale air diffusion simulation and pollutant

forecasting. And the third generation air quality model needs comprehensive and accurate source list and meteorological field data as input to predict, and its application is limited.

With the development of deep learning, AQP methods based on ML have attracted more attention Yi et al. (2018); Zou et al. (2021). The AQP method based on ML takes advantage of a large number of historical observation data for training and testing, finding out the change law of pollutant concentration, and then predicts the air quality, which include linear statistical models Moisan et al. (2018), fitting optimization techniques Niu et al. (2017), and deep learning methods Ma et al. (2020). Since deep learning has a powerful function to automatically extract nonlinear features, recent literatures about AQP often rely on deep learning models. Zhang et al. Zhang et al. (2016) proposed a deep learning AQP method combining CNN (Convolutional Neural Network) and LSTM (Long Short Term Memory), which achieved good results and made scholars see the dawn of the application of deep learning in AQP. Subsequently, Du et al. Du et al. (2019) used one-dimensional CNN to capture local time trend and bidirectional LSTM to extract long-term time series features, and then to construct a hybrid neural network for AQP. Liang et al. Liang et al. (2018) proposed a GeoMAN based on LSTM and encoder-decoder architecture for AQP, and took advantage of the attention mechanism to capture the spatial impact relationship among different stations. Yi et al. Yi et al. (2018) proposed the DeepAir model to construct a sub-network with multi-source data, and used the fusion network to integrate the results of different sub-networks to obtain the final predicted value. The above methods captured temporal correlation well by LSTM, but their capture of spatial relationship was obviously insufficient. Although CNN can be used to establish spatial relationships, it is a static spatial relationship, and the distance among stations is fixed. Due to the influence of weather and terrain, the spatial relationship among stations is not a simple static distance relationship, but a dynamic. In addition, the stations in the city are unevenly distributed and sparse, so interpolation is required in the construction of CNN, resulting in a large number of virtual stations, which will affect the forecasting results.

In contrast, graph-based models naturally sidestep the above issue since they shape the concentration values into graph nodes and keep their original distributions in graph structures. Because graph has the ability to construct non Euclidean entity distribution, it can capture spatial relationships well. Thus, in order to compensate for the lack of spatial relationship learning in the above methods, the methods based on GNN (Graph Neural Network) are applied to AQP. Qi et al. Qi et al. (2019) used GNN to learn the spatial relationship among stations, and LSTM to learn the time correlation of stations, so as to build a comprehensive forecasting model GC-LSTM. Lin et al. Lin et al. (2018) used diffusion convolution operation to replace matrix multiplication in GRU (Gate Recurrent Unit) for sequence modeling, and combined with graph convolution operation to build GC-DCRNN for AQP. Xu et al. Xu et al. (2020) proposed ST-MFGCN for AQP. Its main innovation is to obtain the spatiotemporal variation law of vehicle emissions by building a graph structure traffic network, and then predict traffic pollution emissions. Xu et al. Xu et al. (2021) proposed HighAir, i.e., a hierarchical graph neural network-based AQP method, which adopted an encoder-decoder architecture and considered complex air quality influencing factors, e.g., weather and land usage.

The above methods based on GNN use graph structure to effectively construct the spatial relationship among stations, but they fail to fully construct the edges in the graph structure. For example, literature Xu et al. (2021) simply used distance and wind direction similarity to construct edge weights. In order to better build the edge relationship, Wang et al. Wang et al. (2020) identified a set of critical domain knowledge for PM2.5 forecasting and developed a novel graph based model, PM2.5-GNN, they used domain knowledge (Wind speed and directiondistance, advection coefficient) to construct edge weights. Although they make use of domain knowledge, they simply list some impact factors, which is not enough. The pollution impact relationship among stations is comprehensively determined by meteorological conditions (wind speed, wind direction, pressure, humidity, planetary boundary layer height, etc.) and landform. It is a complex process and needs to be analyzed by using professional models.

According to the existing research status, we can know that machine learning is a novel AQP method based on data-driven strategy, which can effectively capture the regularity and time periodicity of pollutant concentration changes and has the characteristics of simplicity, flexibility and rapid deployment. However, these methods are unable to timely capture the regional impact and sudden change caused by the pollutant diffusion in the air. On the other hand, the traditional mechanism model based on the principle of atmospheric diffusion can better simulate the impact of surrounding areas on the forecasting points. Motivated by the above considerations, we use the LSTM (Long Short

Table 1: Quarter feature representing method

|  | Spring | Summer | Fall | Winter |
|---|---|---|---|---|
| spring season | 1 | 0 | 0 | 0 |
| summertime | 0 | 1 | 0 | 0 |
| autumn | 0 | 0 | 1 | 0 |
| wintertime | 0 | 0 | 0 | 1 |

Term Memory) model to learn the timing and periodicity of pollutant concentration, and use the GN-N (Graph Neural Network) model to learn the spatial relationship of pollutants among stations, and thus building a multi granularity spatiotemporal graph neural network model, called MGST-GNN. In order to better capture the mutation of pollutant concentration and its impact on air diffusion and deposition, the mechanism model HYSLPLIT is used to dynamically construct the adjacency matrix and edge relation of the spatiotemporal graph. The main contributions of this paper include:

- We innovatively propose a dynamic spatiotemporal graph model combining mechanism model and graph neural network. The adjacency matrix and edge weight vector of dynamic graph are constructed based on the simulation results of diffusion, transport and deposition of polluted air mass by mechanism model, so that the architecture learns the spatial influence relationship among multi granularity stations.

- In order to better learn the temporal patterns and periodicity of pollutants, we propose to add time characteristic attributes of quarter, month, week, hour and holiday to each node in the encoder, use LSTM based on attention mechanism for temporal learning in the decoder to enhance MGST-GNN and use a new data augmented method to solve the problem of insufficient training data.

## 2 PRELIMINARY

### 2.1 RELATED CONCEPTS AND PROBLEM DEFINITION

**Definition 1 Region and Stations:** we set $R = \{r_a, 1 \leq a \leq N\}$ as a set of $N$ regions, $L = \{l_a, 1 \leq a \leq N\}$ as a location set of $N$ regions, $S_a = \{s_{a,i}, 1 \leq i \leq |S_a|\}$ as a set of stations in region $r_a$, $L_a = \{l_{a,i}, 1 \leq i \leq |S_a|\}$ as a location set of stations in region $r_a$, where $l_{a,i}$ is composed of longitude and latitude of a station, and $l_a$ is the mean value of $l_{a,i}$ in $L_a$. Note: According to different application scenarios, the region referred to in this paper can be a national region, a city region or an administrative region, etc.

**Definition 2 Time Feature (TF):** TF includes five features: quarter, month, week, hour and holiday. We represent the quarter by using one-hot encoding adopted for 4-bit binary representation, as shown in Table 1.

Similarly, the month is represented with 12-bit binary by one-hot encoding; week is represented with 7-bit binary by one-hot encoding; Hour is represented with 24-bit binary code by one-hot coding mode; Holiday is represented with 2-bit binary by one-hot coding (0 means non-holidays, 1 means holiday).

**Definition 3 Geomorphic Feature (GF):** GF contains the topography and land usage information of a station, which consider the altitude and five land usage categories: residential area, park, mountain, water (river or pool), and industry. Where, the altitude is divided into four categories: very high (more than 1300m), high (1000-1300m), medium (500-1000m) and low (less than 500m), and the corresponding categories are represented by the numbers 1, 2, 3 and 4; The land usage type is determined by the number of major land use types within 10 km around the station. For example, the altitude of $s_{a,i}$ is 800 meters, and there is one residential area, two park, one mountain, three pool, and two industrial facilities within the perception radius 10 kilometers of $s_{a,i}$. Thus, the GF vector $gf_{a,i}$ can be represented as $[3, 1, 2, 1, 3, 2]$.

**Definition 4 Weather Data (WD):** The WD of region $r_a$ and stations stations $s_{a,i}$ at time slot $t$ are represented as the vectors $wd_a^t$ and $wd_{a,i}^t$, including temperature, humidity, rainfall, wind speed, wind direction, and air pressure.

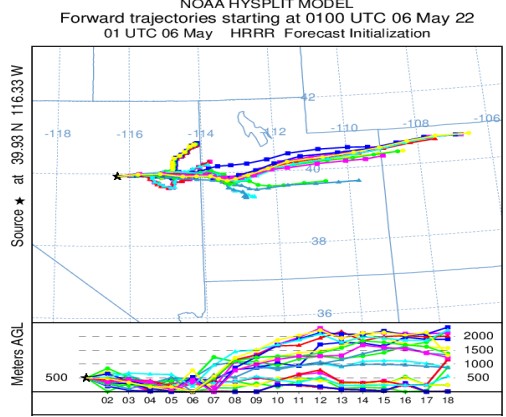

Figure 1: Trajectory analysis chart (Different colored lines in the figure represent trajectories of different heights. The points on each line represent the location of the trajectory, which is represented by latitude and longitude.)

**Definition 5 Pollutant Concentration Data:**  The pollutant concentration data includes the concentrations of six major pollutants such as PM2.5, PM10, $SO_2$, $CO_2$, CO and $O_3$. Among them, PM2.5 and $O_3$ are the most concerned at present. Therefore, the later experiments focus on the concentration prediction of these two pollutants.

**Definition 6 HYSPLIT:** HYSPLIT Stein et al. (2015); Warner (2018) is a complete system for computing simple air parcel trajectories, as well as complex transport, dispersion, chemical transformation, and deposition simulations. HYSPLIT continues to be one of the most extensively used atmospheric transport and dispersion models in the atmospheric sciences community. A common application is a back trajectory analysis to determine the origin of air masses and establish source-receptor relationships. HYSPLIT has also been used in a variety of simulations describing the atmospheric transport, dispersion, and deposition of pollutants and hazardous materials.

In this paper, HYSPLIT is used to establish the source-receptor relationship among nodes by trajectory analysis. When HYSPLIT is used for trajectory analysis, it only needs to input the meteorological data of the simulation area and the coordinate information of the initial point of the simulation. The meteorological data can be downloaded from the official website of Air Resources Laboratory (ARL) [1] for free. Fig. 1 shows an example of trajectory analysis using HYSPLIT. The example takes Beijing Center $(116°20', 39°56')$ as the starting point and 10:00 on May 6, 2022 as the starting time to predict the air mass trajectory in the next 48 hours. It can be seen from Fig. 1 that the location and time of each track can be obtained by HYSPLIT.

## 2.2 PROBLEM STATEMENT

AQP task: Given region locations $L$, station locations L, geomorphic feature data $gf_{a,i}$, $\tau_{in}$ hours of pollutant concentration data $con$, and $\tau_{in} + \tau_{out}$ hours of weather data $wd$, the AQP task aims to forecast the pollutant concentrations of stations for the next $\tau_{out}$ hours, where $\tau_{in}$ denotes the length of historical time window and $\tau_{out}$ denotes forecasting horizon.

## 3 METHODOLOGY

### 3.1 AQP MODEL FRAMEWORK

Existing AQP methods based on GNN Du et al. (2019); Lin et al. (2018) mostly simulate the spatial relations among stations by constructing a flat static graph. However, due to the distance among stations and terrain, it is difficult to transfer information among stations in different regions through flat graph. In addition, the spatial relationship among stations is greatly affected by the dynamic wind

---

[1]https://nomads.ncep.noaa.gov/pub/data/nccf/com/hysplit

direction, wind speed and other meteorological parameters, so it is difficult to construct its edge relationship by means of static graph. Finally, pollutant concentrations at different time in history have different impacts on current air quality. In order to overcome the shortcomings of flat static graphs, we build an encoder-decoder architecture based on dynamic multi-granularity spatiotemporal graph by referring to literature Wang et al. (2020), as shown in Figure 2. In the encoding stage, multi-granularity graph network is used to learn the spatial relationship among stations and LSTM network is used to learn its temporal relationship. In the decoding stage, auxiliary data and attention mechanism are used to enhance LSTM forecasting learning and decode the future pollutant concentration value. This framework fully considers three key factors affecting air quality, namely, meteorology, space and time. Among them, the multi-granularity spatiotemporal graph neural network (as shown in Figure 3) is the focus.

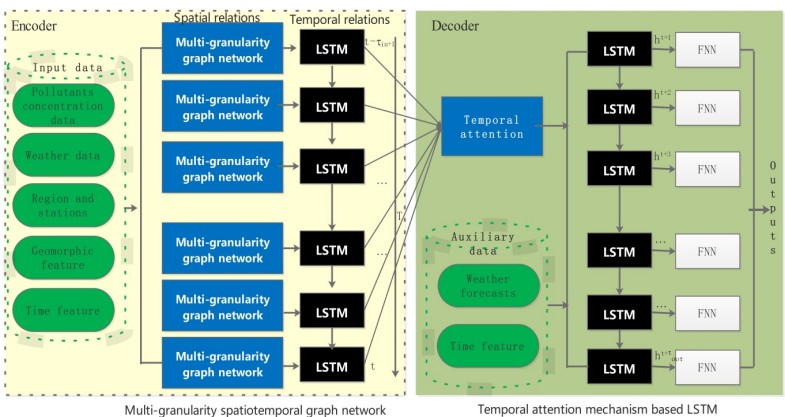

Figure 2: Framework diagram of AQP model

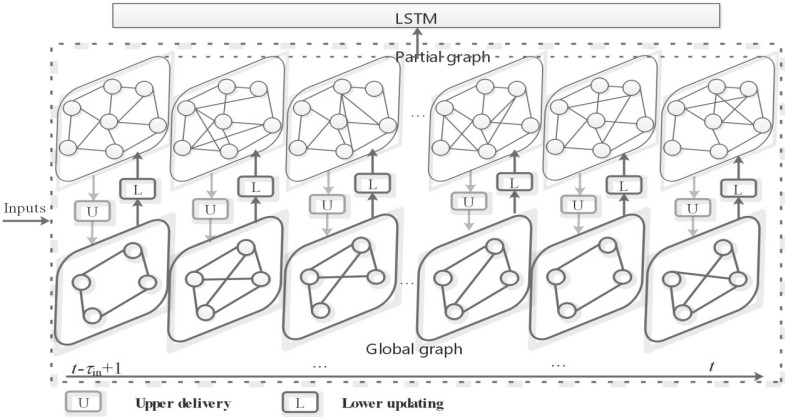

Figure 3: Multi-granularity spatiotemporal graph neural network

## 3.2 SPATIOTEMPORAL GRAPH CONSTRUCTION

We propose a multi-granularity spatiotemporal graph network (MGST_GNN) by referring to reference Xu et al. (2021). Assume that $V$ is the set of nodes, $E$ is the set of connected edges among nodes, and $u$ is the global attribute of a station-level partial subgraph. Global graph and partial subgraph interact through message passing mechanism. Partial subgraph's information is upper delivered to the global graph; the global graph's information is transmitted downward to the partial subgraph. In each time period, MGST_GNN will calculate the attribute information of each global node and use it to update the corresponding partial node attribute. For each region of the partial subgraph, we calculate the overall air quality information representation of the regions at each time slot,

forming a sequence sent to the LSTM to get the representations of current and historical pollutant concentration.

Taking the AQP of all stations in a city as an example, all stations in the city are clustered according to their longitude and latitude information to obtain $N$ categories, that is, $N$ regions are clustered. Therefore, each region can be regarded as a node in the global graph, while each station in the region can be regarded as a node in the partial subgraph. A graph composed of region-level nodes is called a global graph, while a graph composed of station-level nodes is called a partial subgraph. From this, a multi-granularity graph network can be constructed, as shown in Figure 4, where the coarse-grained graph is a region-level global graph and the fine-grained graph is a station-level partial subgraph.

### 3.2.1 ATTRIBUTE DEFINITION OF GRAPH NODE

To the node in the global graph, its attribute is the pollutant concentration value of the node region. To the node in partial subgraph, its attribute is composed of the pollutant concentration value of the node station, GF and TF.

### 3.2.2 EDGE AND ATTRIBUTE DEFINITIONS OF GRAPHS

Wind speed, wind direction, rainfall and other meteorological data have a decisive influence on the horizontal transmission of pollutants. In order to make use of this domain knowledge, we adopt the air quality model HYSPLIT to learn the relationship among nodes and build dynamic connection edges. Taking the edge calculation of global graph as an example, the city is divided into grids according to the clustered regions, so that each region falls into a unique grid. For each region (node $v_a$), HYSPLIT is used to calculate all trajectories starting from $v_a$ and stepping in hours within the next time $t$. With HYSPLIT, we can track the time when each trajectories crosses the grid where the node is located, and record the number of pass trajectories and crossing time of the grid from other nodes except $v_a$, and dynamically construct the global connection edge and its attribute vector. Repeating these steps until each station is analyzed as a starting point. For example, taking $v_1$ as the starting point, using HYSPLIT to calculate that there are 5 trajectories passing through $v_2$ in the following 48 hours, and the time interval is 1 hour, 3 hours, 4 hours, 8 hours and 48 hours in the future, then the edge relationship between node $v_1$ and $v_2$ is formed, and its attribute vector is: $e_{12} = [1, 0, 1, 1, 0, 0, 0, 1, 0, 0, 0, 0, 0, 0, 0, 0, 0, 0, 0, 0, 0, 0, 0, 0, 0, 0, 0, 0, 0, 0, 0, 0, 0, 0, 0, 0, 0, 0, 0, 0, 0, 0, 0, 0, 0, 0, 0, 1]$. From $e_{12}$, we can get the specific impact time of node $v_1$ to $v_2$. It is a dynamic, specific and accurate edge attribute, which is calculated by the mechanism model according to the meteorological conditions and geographical location. Station-level partial subgraph edge and attribute calculation are similar.

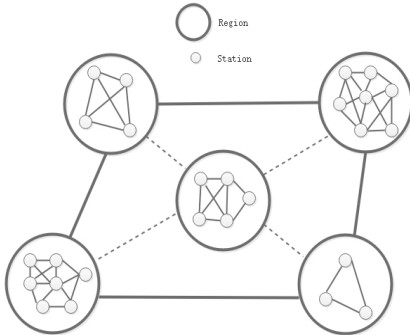

Figure 4: Multi-granularity graph structure

### 3.3 MODELING OF SPATIAL DEPENDENCIES

MGST_GNN can model spatial dependencies with different granularities by message passing mechanism, which mainly includes message aggregation and description updating. Message aggregation is shown as follows:

$$M_a = \{(x_s, x_a, e_{a,s})\}_{s \in N(a)} \tag{1}$$

$$m_a \leftarrow \Psi(M_a) \tag{2}$$

where $M_a$ denotes the set of all the messages passed to node $v_a$; $x_a$ is the attribute of node $v_a$; $x_s$ is the attribute of a neighbor node; $e_{a,s}$ is the edge attribute; $N(a)$ denotes the neighbor node net of node $v_a$ (dynamically determined by HYSPLIT trajectory); $m_a$ is the aggregation vector of node $v_a$; $\Psi(\cdot)$ denotes aggregate function.

Description updating is shown as follows

$$\begin{cases} x_a^{'} \leftarrow \phi_1(m_a, x_a), & \text{in global graph} \\ x_{a,i}^{'} \leftarrow \phi_2(m_{a,i}, x_{a,i}, u_a), & \text{in partial graph} \end{cases} \tag{3}$$

where $\phi_1$ and $\phi_2$ denote update functions; $m_{a,i}$ is the aggregation vector of node $v_{a,i}$ in the $a$-th partial subgraph; $x_{a,i}$ is the attribute of node $v_{a,i}$; $u_a$ is the global attribute of $a$-th partial subgraph; $\phi_1$ and $\phi_2$ can be implemented by using different FNNs.

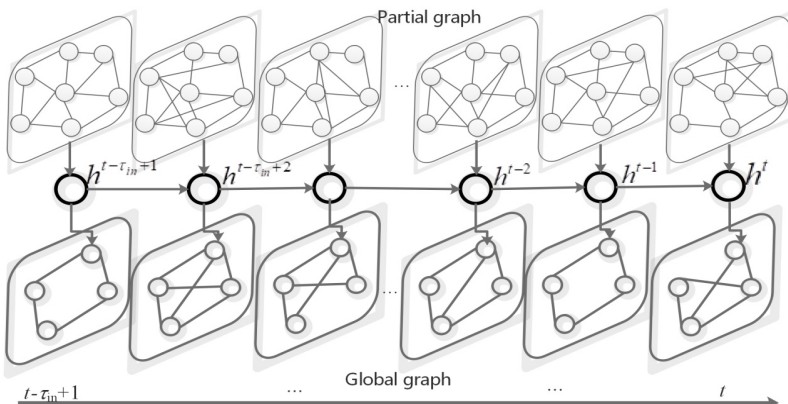

Figure 5: Message aggregation diagram

The specific message aggregation method is from the partial subgraph to the global graph, as shown in Figure 5. The information transmitted from the partial subgraph includes the current and historical pollutant concentration values, and the average method is adopted to aggregate the information transmitted from the partial subgraph:

$$con_a^t = \text{mean}(con_{a,i}^t), \quad 1 \le i \le |S_a| \tag{4}$$

where $con_a^t$ denotes the aggregate value at time $t$ of node $v_a$ in the global graph; $con_{a,i}^t$ denotes the pollutant concentration value at time $t$ of node $v_{a,i}$ in the $a$-th partial subgraph; $|S_a|$ is the number of nodes in the $a$-th partial subgraph.

For nodes in the global graph, message aggregation is used to calculate the global representation of pollutant concentration in each time slot, thus forming a series of sequences $\{con_a^{t-\tau_{in}+1}, con_a^{t-\tau_{in}+2}, \ldots, con_a^t \mid 1 \le a \le N\}$. These sequences are then fed into the global LSTM to learn the current and historical representations. The notation $h$ in Figure 5 represents the hidden layer state of each time slot in LSTM, and $h$ is taken as the initial attribute of nodes in the global graph, so the global graph includes current and historical information.

The specific updating method is to update the partial subgraph by using the global graph, as shown in Figure 6. The information transmitted from the global graph includes the historical pollutant concentration values of all regions. At each time slot, the output of the global graph is fed into the FNN to obtain a downward update vector, which is used to update the global attribute $u$ of the partial subgraph. The $u$ includes meteorological parameters and downward update vector, which are used

in the partial subgraph by message passing. Therefore, the nodes in the station-level partial subgraph can make use of the historical information of neighbor nodes in the region-level global graph.

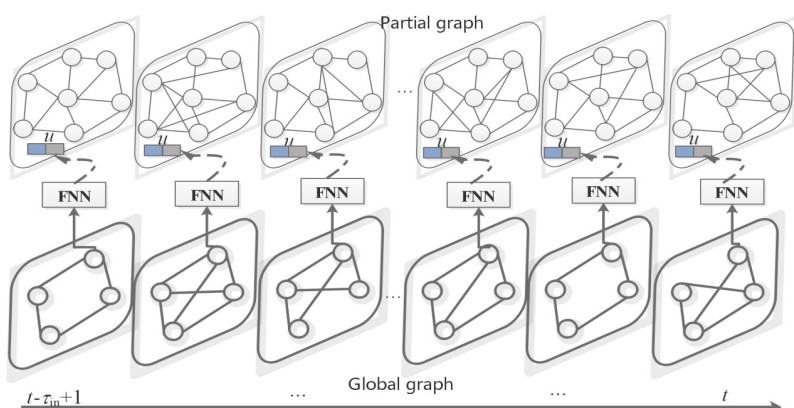

Figure 6: Description updating chart

## 3.4 MODELING OF TEMPORAL DEPENDENCIES

In order to capture the temporal dependence, we adopt an encoder-decoder architecture, as shown in Figure 1. In each time slot, node attributes in the partial subgraph form a sequence $X_{a,i} = \{x_{a,i}^{t-\tau_{in}+1}, x_{a,i}^{t-\tau_{in}+2}, \ldots, x_{a,i}^{t} \mid 1 \le a \le N, 1 \le i \le |S_a|\}$. That is, each time slot in the historical has a corresponding node attribute. The LSTM in the encoder uses $X_{a,i}$ as the input and the final state of the LSTM is used as the input of the decoder. The input of the LSTM in the decoder includes not only the output of the encoder, but also the node attribute in the partial subgraph. The output of LSTM in the decoder is used as the input of FNN, and FNN outputs the predicted pollutant concentration value in the future $\tau_{out}$ time slot.

In order to make better use of temporal characteristics, temporal attention mechanism is introduced in the decoding stage to learn the dynamic temporal correlation between future time and historical time. Give the hidden state $h'_{t'-1}$ and cell state $c'_{t'-1}$ of the LSTM in the decoder at time $t'-1$, then at time $t'$, the attention weight of the hidden state $h_t$ output by the encoder is calculated as follows:

$$\rho_{t'}^{t} = v^T \tanh(W[h'_{t'-1}; c'_{t'-1}] + Uh_t + b) \tag{5}$$

$$\lambda_{t'}^{t} = \frac{\exp(\rho_{t'}^{t})}{\sum_{t=1}^{T} \exp(\rho_{t'}^{t})} \tag{6}$$

where $\lambda_{t'}^{t}$ is the attention weight; $v, b, W, U$ are the parameters to be learned. Through the above formulas 5 and 6, the attention weight of all historical hidden states in the encoder can be calculated, and then, the hidden state $h_t$ is weighted and summed to obtain the time context vector $c$

$$c_{t'} = \sum_{t=1}^{T} \lambda_{t'}^{t} h_t \tag{7}$$

The output result $\hat{o}_{t'-1}$ of decoder at time $t'-1$, the meteorological data $wd_{t'}$ at time $t'$, the time feature $tf_{t'}$ and the time context vector $c_{t'}$ are connected as the input for the LSTM of the decoder at time $t'$, and it is used to update the hidden state $h'_{t'}$:

$$h'_{t'} = \text{LSTM}(h'_{t'-1}, [\hat{o}_{t'-1}; wd_{t'}; tf_{t'}; c_{t'}]). \tag{8}$$

## 4 EXPERIMENTS

### 4.1 EXPERIMENTAL DATASETS

#### 4.1.1 JINAN DATABASE (JN)

Jinan database is a city-level data set collected by us in projects. Jinan is located in the middle of Shandong Province, China. Its geographical position is between $36°01'\sim 37°32'$ N and $116°11'\sim 117°44'$ E. It is distributed in a narrow and long terrain with a total area of 7998 square kilometers. There are 130 air monitoring stations in Jinan. Each station outputs the concentration values of pollutants (PM2.5, PM10, $SO_2$, $CO_2$, CO, $O_3$) and meteorological parameters (rainfall, surface pressure, temperature, humidity, wind speed and wind direction) every hour. We collected the historical monitoring data of 130 stations in Jinan from January 1st, 2019, to January 1st, 2022, as the training and test set.

We divide the 130 stations into 13 regions, and the global graph consists of 13 regions. Each region is used as a global node in the global graph. The stations in each region form a partial subgraph, and the stations in the region are the nodes of the partial subgraph.

#### 4.1.2 YANGTZE RIVER DELTA CITY GROUP DATABASE(YRD)

The city group contains ten cities: Shanghai, Hangzhou, Suzhou, Ningbo, Shaoxing, Jiaxing, Wuxi, Zhoushan, Nantong, and Huzhou. We used Air Pollution Prediction system [2] to collect historical pollutant concentration values and meteorological parameters of corresponding stations, and the time span was from January 1th, 2019, to December 31th, 2022. Therefore, we collected 3 years of historical monitoring data as the training and test set. Each city in the Yangtze River Delta city group is a global graph node, and each station in the city is a node of the corresponding partial subgraph.

Geographic features and weather forecast data are obtained by:

- Geographic features are collected from the map engine of AMAP [3].The perception radius is set to 1000 m.
- Weather forecast data are collected from the Air Resources Laboratory (ARL)[4]. This website can download the meteorological forecast data for the following 26 days at most, with an accuracy of $0.25 \times 0.25$ degrees, which can be updated four times a day.

We use min-max normalization to normalize the pollutant concentration and meteorological data into [0, 1].

### 4.2 EXPERIMENTAL SETTINGS

We split the dataset into training data, validation data, and test data by the radio of 0.7:0.1:0.2. We choose Adam Kingma & Ba (2014) as the optimizer in the training phase. During the training phase, the batch size is set to 128 and the epoch size is set to 500, and use RMSprop Xu et al. (2020) for 50 epochs with learning rate as $5^{-4}$. The hidden size of GNNs is set to 32, and the hidden state size of LSTMs is set to 64.

We implement our method by PyTorch Paszke et al. (2019), constructing GNNs with PyTorch geometric library Fey & Lenssen (2019) and implement HYSPLIT trajectory analysis and edge weight construction with PySPLIT Warner (2018). The code is released on GitHub. A server with one CPU (Intel®Xeon®Platinum), and one GPU (NVIDIA Tesla T4) accomplishes all computing tasks, including training, validation, and test.

We introduce two metrics: MAE (mean absolute error) and SMAPE (symmetric mean absolute percentage error), to evaluate the performances of all methods. In the experiments, we utilize previous 48-hour observations and select the results of 1 hour, 6 hours, 12 hours, 18 hours, 24 hours, 36 hours and 48 hours ahead forecasting to report. All experiments are repeated 5 times to avoid contingency.

---

[2]http://airprediction.urban-computing.com
[3]https://lbs.amap.com/api/webservice/guide/api/search/
[4]https://nomads.ncep.noaa.gov/pub/data/

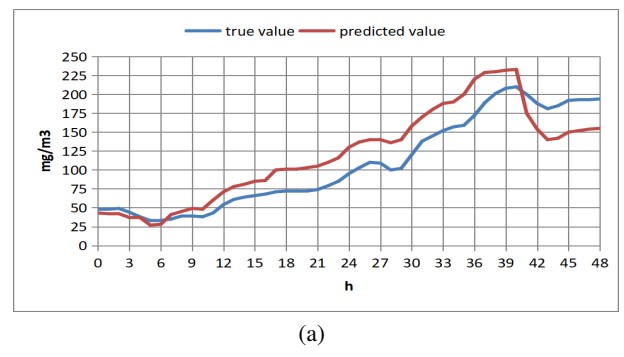

(a)

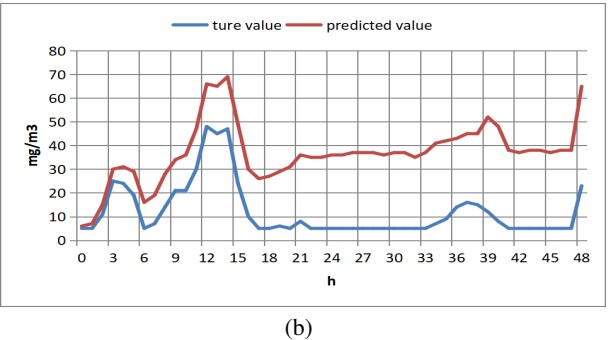

(b)

Figure 7: 48-hour predicted value versus true value (a) PM2.5 (b) O$_3$

Table 2: Results of factor evaluation based on JN database

| Method | Metric | 1h | 6h | 12h | 18h | 24h | 36h | 48h |
|---|---|---|---|---|---|---|---|---|
| MGST_GNN | MAE | **6.15** | **15.24** | **20.56** | **25.62** | **34.83** | **43.98** | 52.06 |
| | SMAPE | **0.07** | **0.10** | **0.14** | **0.19** | **0.26** | **0.33** | **0.46** |
| w/o $wdf$ | MAE | 7.06 | 17.17 | 23.68 | 29.33 | 38.56 | 46.70 | 53.78 |
| | SMAPE | 0.07 | 0.11 | 0.16 | 0.21 | 0.28 | 0.35 | 0.47 |
| w/o $tf$ | MAE | 6.76 | 17.16 | 23.56 | 28.45 | 34.96 | 45.55 | **52.01** |
| | SMAPE | 0.07 | 0.11 | 0.16 | 0.20 | 0.26 | 0.34 | 0.46 |
| w/o $gf$ | MAE | 6.71 | 16.78 | 22.56 | 28.64 | 34.95 | 46.56 | 52.11 |
| | SMAPE | 0.07 | 0.11 | 0.15 | 0.20 | 0.26 | 0.35 | 0.46 |

Figure 7 depicts the predicted PM2.5 and O$_3$ concentrations and their corresponding real values of a station in Jinan city from 00:00 on January $2nd$, 2021 to 23:00 on January $3rd$, 2021 for consecutive 48 hours. From Figure 7, we can see that it is easier to predict O$_3$ concentration than PM2.5, because O$_3$ has more periodic regularity and stability than PM2.5. Therefore, the subsequent experiments were carried out with the forecasting of PM2.5 concentration.

## 4.3 MULTI-SOURCE FACTOR EVALUATION

To verify the effectiveness of multiple factors, we compare MGST_GNN with three variants, each of which removes one kind of factors. Specifically, MGST_GNN w/o $wdf$ removes meteorological forecast data; MGST_GNN w/o $tf$ removes time feature and MGST_GNN w/o $gf$ removes geomorphic feature.

The performances of MGST_GNN and its variants are given in Table 2, as can be seen from the table:

1. MGST_GNN outperforms the other three variants, which indicates that all factors can improve the performance of AQP;

Table 3: Results of model component evaluation based on JN database

| Method | Metric | 1h | 6h | 12h | 18h | 24h | 36h | 48h |
|---|---|---|---|---|---|---|---|---|
| MGST_GNN | MAE | **6.15** | **15.24** | **20.56** | **25.62** | 34.83 | 43.98 | **52.06** |
| | SMAPE | **0.07** | **0.10** | **0.14** | **0.19** | **0.26** | **0.33** | **0.46** |
| w/o multi granularity | MAE | 7.12 | 17.10 | 23.59 | 28.02 | 34.95 | 48.02 | 53.58 |
| | SMAPE | 0.08 | 0.12 | 0.16 | 0.21 | 0.26 | 0.36 | 0.47 |
| w/o HYSPLIT | MAE | 7.16 | 18.06 | 23.59 | 28.16 | **34.76** | 48.35 | 53.02 |
| | SMAPE | 0.08 | 0.13 | 0.16 | 0.21 | 0.26 | 0.36 | 0.47 |
| w/o taLSTM | MAE | 6.15 | 15.85 | 21.56 | 28.64 | 34.96 | **43.16** | 52.25 |
| | SMAPE | 0.07 | 0.10 | 0.15 | 0.21 | 0.26 | 0.33 | 0.46 |

2. The rank of the effectiveness of factors is WD>TF>GF. This result shows that air quality is mostly impacted by weather conditions.

## 4.4 MODEL COMPONENT EVALUATION

To explore the effectiveness of different components, we compare MGST_GNN with three following variants:

1. MGST_GNN w/o multi granularity, which removes the global graph and the corresponding interactions, i.e., the region representation is removed from the global attributes of the partial subgraphs;

2. MGST_GNN w/o HYSPLIT, which removes dynamic edge weight vector by HYSPLIT and use vectors of wind direction and distance instead;

3. MGST_GNN w/o taLSTM, which removes temporal attention mechanism based LSTM at the decoder, and directly use LSTM instead.

The performances of MGST_GNN and the variants mentioned above are given in Table 3, as can be seen from the table:

1. MGST_GNN outperforms MGST_GNN w/o multi granularity. The result indicates that the air quality of adjacent regions is beneficial, which can be used to model the diffusion processes of air pollutants from adjacent regions;

2. MGST_GNN outperforms MGST_GNN w/o HYSPLIT in all metrics. The result indicates that compared with taking wind direction and distance as the edge weight vector, using HYSPLIT to dynamically adjust the weights of edges is a more effective strategy, which can model the effect patterns of wind direction on air pollutant diffusion with domain knowledge;

3. The attention mechanism based LSTM at the decoder is helpful for longer-term prediction, but for short-term prediction.

## 4.5 COMPARISON WITH OTHER PREDICTION METHODS

In order to verify the advanced nature of our method, we compare MGST_GNN with the following prediction methods:

1. HighAir Xu et al. (2021): a GNN network-based air quality prediction method , which adopts an encoder-decoder architecture and considers complex air quality influencing factors, e.g., weather and land usage;

2. PM2.5-GNN Wang et al. (2020): a graph based model with identify a set of critical domain knowledge for PM2.5 forecasting;

3. GC-DCRNN Lin et al. (2018): GC-DCRNN combines recurrent neural networks with diffusion convolution to forecast the air quality. The model describes the spatial relationship by constructing a graph based on the similarity of the built environment among the locations of air quality sensors. To keep the fairness of comparison, we introduce weather data into the input of the decoder;

Table 4: Model comparison results based on YRD database

| Method | Metric | 1h | 6h | 12h | 18h | 24h | 36h | 48h |
|--------|--------|-----|------|------|------|------|------|------|
| MGST_GNN | MAE | 7.16 | **17.29** | **21.25** | 27.92 | **36.97** | **45.28** | **54.73** |
| | SMAPE | **0.08** | **0.12** | **0.14** | **0.20** | **0.27** | **0.34** | **0.48** |
| HighAir | MAE | **7.12** | 17.83 | 22.65 | **27.49** | 38.85 | 48.36 | 55.68 |
| | SMAPE | 00.08 | 0.12 | 0.15 | 0.20 | 0.29 | 0.36 | 0.49 |
| PM2.5-GNN | MAE | 7.15 | 18.09 | 23.14 | 29.57 | 37.96 | 48.99 | 56.12 |
| | SMAPE | 0.08 | 0.13 | 0.15 | 0.21 | 0.28 | 0.36 | 0.50 |
| GC-DCRNN | MAE | 7.56 | 18.56 | 24.88 | 30.67 | 43.26 | 51.84 | 60.02 |
| | SMAPE | 0.08 | 0.13 | 0.17 | 0.22 | 0.33 | 0.45 | 0.53 |
| GC-LSTM | MAE | 8.13 | 18.97 | 24.53 | 31.52 | 42.51 | 51.98 | 58.96 |
| | SMAPE | 0.09 | 0.13 | 0.17 | 0.22 | 0.33 | 0.45 | 0.52 |
| ST-UNet | MAE | 7.22 | 18.69 | 23.36 | 27.86 | 40.16 | 51.97 | 57.85 |
| | SMAPE | 0.08 | 0.13 | 0.16 | 0.20 | 0.32 | 0.45 | 0.51 |
| STA-LSTM | MAE | 8.15 | 19.21 | 24.62 | 31.96 | 42.87 | 50.68 | 60.37 |
| | SMAPE | 0.09 | 0.14 | 0.17 | 0.23 | 0.33 | 0.37 | 0.53 |

4. GC-LSTM Qi et al. (2019): GC-LSTM integrates LSTM and Graph Convolutional Networks (GCN) to model the temporal and spatial dependency respectively. Differing from our MGST_GNN, the GCN module in GC-LSTM only applies to undirected graph, and no edges' attributes are used;

5. ST-UNet Yu et al. (2019): ST-UNet is a spatial-temporal prediction method that leverages pooling operation to coarsen a graph in spatial domain and adopts dilated RNN to capture temporal dependencies;

6. STA-LSTM Zou et al. (2021): STA-LSTM is a long short-term memory air quality prediction model based on a spatiotemporal attention mechanism.

The performances of MGST_GNN and other prediction methods are given in Table 4, as can be seen from the table:

1. MGST_GNN outperforms GC-DCRNN, GC-LSTM and ST_UNet, especially in long-term forecasting. It is indicated that a multi-granularity structure can model spatial dependencies more effectively than a flat structure. The reason is that the multi granularity graph not only considers the local impact of neighboring stations on the prediction station, but also the global impact of different regions on the prediction station, which makes the development of spatial relationships more sufficient;

2. MGST_GNN outperforms HighAir. This is because in the aspect of spatial relationship learning, MGST_GNN uses the professional HYSPLIT model to build edge weight vector. It comprehensively uses meteorological and topographical conditions to calculate the influence relationship among nodes and the specific influence time, making the construction of spatial relationship more accurate and delicate. HighAir simply uses wind direction and distance to construct rough edge weight vector. In terms of temporal relation learning, MGST_GNN adds time feature, and uses the attention mechanism based LSTM in decoder. HighAir simply uses LSTM to build temporal relationships;

3. MGST_GNN outperforms PM2.5-GNN. This is because MGST_GNN uses a multi-granularity spatiotemporal graph network and combines meteorological data, topographic features, time features and professional models to construct node and edge attributes. However, PM2.5-GNN is only a single-granularity spatiotemporal graph network. In the construction of edge attributes, although they make use of domain knowledge, they simply list part of the influencing factors, which is still insufficient. When building node attributes, they only use meteorological data.

## 5 CONCLUSION

In this paper, the influence of meteorological, spatial and temporal factors on AQP is fully considered, and an encoder-decoder architecture based on multi-granularity spatiotemporal graph network

is proposed to predict pollutant concentration over a long period of time. Compared with the existing models, the striking characteristic of this paper is that the meteorological, spatial terrain and time factors are considered comprehensively through the professional air quality model, while other models take the influencing factors as the splitting parameter input. That is, the model in this paper integrates the advantages of mechanism model and machine learning, and namely it is a comprehensive model. The experimental results show that the proposed model is of progressiveness and has good applicability.

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

# A APPENDIX

## A.1 DATA AUGMENTATION AND EXPERIMENTATION

Aiming at the difficulty of collecting training samples for air quality prediction, we use the method of data augmentation to reconstruct samples. Due to the distribution difference between real weather data and forecast weather data, Gaussian noise is introduced to the meteorological data in the sample to enhance the data and improve the generalization ability of the model. In addition, due to the influence of monitoring instruments, environment and other factors, the pollutant concentration values collected may be biased. Therefore, the pollutant concentration values of samples are disturbed up and down by 1 metric to enhance the number of samples and improve the model generalization

Table 5: List of sample augmentation

| Augmentation type / Sample type | Meteorological data plus noise | Concentration value disturbance |
|---|---|---|
| original sample | - | - |
| sample 1 | ✓ | - |
| sample 2 | - | ✓ |
| sample 3 | ✓ | ✓ |

Table 6: MAE results of data augmentation evaluation based on JN database

| Method | Pollutant category | 1h | 6h | 12h | 18h | 24h | 36h | 48h |
|---|---|---|---|---|---|---|---|---|
| MGST_GNN | PM2.5 | **6.15** | **15.24** | **20.56** | **25.62** | **34.83** | **43.98** | **52.06** |
| | $O_3$ | **5.06** | **13.36** | **17.14** | **22.16** | **29.90** | **37.93** | **44.61** |
| w/o DA | PM2.5 | 6.25 | 16.97 | 22.18 | 28.23 | 36.96 | 46.31 | 54.69 |
| | $O_3$ | 5.11 | 14.84 | 18.86 | 23.72 | 30.78 | 39.19 | 48.07 |

ability. Therefore, Gaussian noise is introduced into meteorological data and random perturbation is for pollutant concentration data to enhance the samples by three times. The specific augmentation methods are shown in Table 5. Finally, 26136 samples were obtained from JN and YRD respectively.

In order to verify the data augmentation effect, we conduct a comparison experiment between data augmentation and non-data augmentation, as shown in Table 6, where w/o DA indicates that data augmentation technology is not used. As seen from Table 6: using our data augmentation method, the effect is effectively improved.

### A.2 ABLATION STUDY AND EXPERIMENTATION

In this paper, we are the first to use the professional model HYSPLIT to build the graph dynamically. Therefore, this appendix section will demonstrate the effectiveness of using HYSPLIT to construction graph dynamically through ablation experiments. Specifically, the ablation experiment was conducted based on the current advanced spatiotemporal graph neural network HighAir and PM2.5-GNN. In HighAir, it uses the distance among nodes to statically construct the edge of the graph, and uses the wind direction information between nodes to calculate the edge attribute vector. Therefore, we use HYSPLIT instead of the graph construction method in HighAir. In PM2.5-GNN, it uses the distance between nodes and the altitude of the position to statically construct the edge of the graph, and uses the parameter of domain knowledge between nodes to calculate the edge attribute vector. Therefore, we use HYSPLIT instead of the graph construction method in PM2.5-GNN. The experimental results are shown in Table 7, where HighAir_HYSPLIT and PM2.5-GNN_HYSPLIT respectively indicate that HYSPLIT is used to replace the original graph construction. As can be seen from the table, HYSPLIT builds dynamic graphs better than static graphs of HighAir and PM2.5-GNN.

In order to further verify the effectiveness of using HYSPLIT to dynamically construct graph, especially its advantages for air quality prediction in complex scenarios such as abrupt change in pollutant concentration. We develop a Dataset-mini, where we focus on heating season (November to February). Dataset-mini is more challenging for two reasons. Firstly, during winters, heating emissions can dramatically increase the frequency of air pollution occurrence. Secondly, the direction of prevailing wind is north or northwest, which contributes to pollutant's long-distance transport from North China to South China. The results in Table 8 show that using HYSPLIT to dynamically construct graph can significantly improve the accuracy of the model's air quality prediction on sudden changes in pollution and regional impacts caused by strong winds. This method eliminates the construction process of specially designed auxiliary network to learn the edges of graph and provides a new way for the construction of graph neural network, which can be easily extended to other spatiotemporal forecasting tasks. For example, in water quality prediction, a professional hydrodynamic model (MIKE) can be used to dynamically construct the graph structure, so as to better learn the influence of water quality in different regions on the prediction points.

Table 7: Ablation experiment result based on YRD database

| Method | Metric | 1h | 6h | 12h | 18h | 24h | 36h | 48h |
|---|---|---|---|---|---|---|---|---|
| HighAir | MAE | 7.12 | 17.83 | 22.65 | 27.49 | 38.85 | 48.36 | 55.68 |
| | SMAPE | 0.08 | 0.12 | 0.15 | 0.20 | 0.29 | 0.36 | 0.49 |
| HighAir_HYSPLIT | MAE | 7.12 | 17.46 | 22.18 | 27.50 | 37.65 | 47.38 | 54.75 |
| | SMAPE | 0.08 | 0.12 | 0.15 | 0.20 | 0.28 | 0.35 | 0.48 |
| PM2.5-GNN | MAE | 7.15 | 18.09 | 23.14 | 29.57 | 37.96 | 48.99 | 56.12 |
| | SMAPE | 0.08 | 0.13 | 0.15 | 0.21 | 0.28 | 0.36 | 0.50 |
| PM2.5-GNN_HYSPLIT | MAE | 7.14 | 18.01 | 22.79 | 29.05 | 36.97 | 47.95 | 55.52 |
| | SMAPE | 0.08 | 0.13 | 0.15 | 0.21 | 0.27 | 0.35 | 0.49 |

Table 8: Ablation experiment result based on Data-mini

| Method | Metric | 1h | 6h | 12h | 18h | 24h | 36h | 48h |
|---|---|---|---|---|---|---|---|---|
| HighAir | MAE | 8.02 | 20.14 | 27.55 | 35.27 | 44.18 | 56.46 | 65.47 |
| | SMAPE | 0.08 | 0.14 | 0.20 | 0.27 | 0.33 | 0.50 | 0.59 |
| HighAir_HYSPLIT | MAE | 7.86 | 18.73 | 25.17 | 32.94 | 40.49 | 50.97 | 56.17 |
| | SMAPE | 0.08 | 0.13 | 0.18 | 0.25 | 0.30 | 0.37 | 0.51 |
| PM2.5-GNN | MAE | 8.11 | 21.30 | 27.96 | 34.66 | 43.88 | 57.69 | 67.08 |
| | SMAPE | 0.09 | 0.14 | 0.20 | 0.27 | 0.33 | 0.51 | 0.60 |
| PM2.5-GNN_HYSPLIT | MAE | 7.95 | 19.06 | 26.14 | 32.99 | 40.85 | 50.89 | 55.78 |
| | SMAPE | 0.08 | 0.13 | 0.19 | 0.25 | 0.30 | 0.37 | 0.51 |

