# OpenReview forum: "A spatiotemporal graph neural network with multi granularity for air quality prediction"
_ICLR.cc/2023/Conference — Submitted to ICLR 2023_

### Official Review · Reviewer_VRie · 2022-10-24

**Confidence:** 2
**Correctness:** 3
**Technical Novelty And Significance:** 2
**Empirical Novelty And Significance:** 2
**Recommendation:** 3

**Clarity, Quality, Novelty And Reproducibility:**

I think in its current form, this paper has quite a lot of improvement space for its clarity and quality. It seems that both codes and datasets are not open sourced, I'm not confident about its reproducibility.

**Strength And Weaknesses:**

Strength:
1. The proposed model is easy to understand and the idea behind partial graph and global graph is explained well.
2. The experimental results contain many ablation study.

Weaknesses:

1. The contents of this paper definitely can be organized better.
* The experimental results are all in appendixes. The experiments section in main body only have some vague description of the model's performance, e.g. "the MAE ... decreases about 1.88 and 1.52 ...", or "MAE decreases about 1.99, 2.09, 0.59...". This is confusing and it's hard for a general reader to understand what these metrics mean unless they go to check the tables in the appendix. I'd suggest to make the paper's main body self-contained.
* The authors listed many related models in the related work section. However, the differences between existing models and the proposed model were not discussed thoroughly.

2. The experimental results were not presented well. For table 6 in appendix A.6, the first two rows (method MGST_GNN) are all bolded. However, it doesn't seem that MGST_GNN gets the best metric in each column. For example, under metric=1h, HighAir's MAE is 7.12 while MGST_GNN's is 7.16. Under metric=18h, MGST_GNN's MAE is 27.92 while HighAir's is 27.49 and ST-UNet's is 27.86. I think the tradition is to bold the best metric in the corresponding column.

3. In Appendix A.6 the authors claimed that MGST_GNN outperforms the baseline models and provided their explanations. I don't think this is very convincing. Even if MGST_GNN gets better results in Table 6, it doesn't imply that, for example, MGST_GNN outperforms HighAir model because MGST_GNN uses HYSPLIT model. A more promising way is to provide ablation study results on the same dataset.

4. The proposed model uses HYSPLIT to construct the graphs. Since HYSPLIT is specifically designed for air quality related tasks, how does MGST_GNN generalize to other spatiotemporal forecasting tasks? It would be good to see some discussions along this line.

**Summary Of The Paper:**

This paper proposes a MGST_GNN model which uses multi-granularity graph neural network and LSTM to do air quality forecasting. In the air quality forecasting task, a region contains multiple air quality stations. The authors define two graphs: The global graph has nodes as regions and the partial graph (within one region) has nodes as the stations in that region. The authors propose a new message passing model that incorporates these two kinds of graphs. The proposed model uses LSTM to learn the temporal dependencies. The authors conducted experiments on two real world datasets and showed the proposed method achieve good results.

**Summary Of The Review:**

There are a lot of improvements that can be made in this paper's organizations, experimental results presentations and discussions. I think it's not ready yet.

---

### Official Review · Reviewer_mQij · 2022-10-24

**Confidence:** 4
**Correctness:** 3
**Technical Novelty And Significance:** 2
**Empirical Novelty And Significance:** 2
**Recommendation:** 3

**Clarity, Quality, Novelty And Reproducibility:**

The writing is easy to follow. But the novelty is limited. We have concerns about the reproducibility, because the hyperparameter setting is not provided.

**Strength And Weaknesses:**

Strength:
1. The topic of air quality prediction is of great social impact.

Weaknesses:
1. The format of the paper is unprofessional. All the experimental results are in the appendix.
2. Figures are in poor resolution. Please use vectorized images for professional academic writing.
3. The technical contribution is limited. The settings of LSTM, GNN, and attention mechanism are commonly used for air quality prediction. The motivation of multi-granularity also has been studied thoroughly for spatial-temporal forecasting.
4. The experiment cannot fully support the validity of the model. Air quality prediction is inherently a time series prediction task. Time series forecasting models, such as N-BEATS, autoformer, informer, etc, are expected to compare.

**Summary Of The Paper:**

This paper provides a method to predict air quality by combining spatial-temporal data and graph neural networks. The contributions of this paper are mainly on integrating HYSPLIT for modeling the dynamic relationships between nodes.

**Summary Of The Review:**

The topic is important. But the novelty is limited, the experiment is weak. The paper needs lots of effort to improve.

---

### Official Review · Reviewer_tjqh · 2022-10-24

**Confidence:** 2
**Correctness:** 3
**Technical Novelty And Significance:** 2
**Empirical Novelty And Significance:** 2
**Recommendation:** 6

**Clarity, Quality, Novelty And Reproducibility:**

The techniques used in this paper are solid.
The overall presentation is good and clear.
The originality is average.


**Strength And Weaknesses:**

Strength
1 The paper addresses an important problem in air pollutant concentration prediction. By integrating the mechanical model and machine learning, the work bridges the traditional methods and current trend.
2 The high-level presentation is good which is easy to follow and understand.
3 The experiment is solid and comprehensive.
Weakness
1 The paper uses inconsistent terms and symbols, which is confusing at some time
For example, R and r_a are to represent the region in section 3.1, but authors use the same symbol for messages in equation 2, which is a little ambiguous. In figure2, the input data doesn’t correspond to the variables in the definition section, I would suggest the authors to use consistent symbolic notations for the important features in the both the text and the explanatory figures. Some of the figures need more explanation, e.g., in figure1, the meanings of the axes are unknown. Typo: neighbor node net->neighbor node set
2 The novelty is not outstanding. Although integrating the mechanical model and machine learning is interesting, most of the techniques are well-established and the authors simply apply them to the new problem.  The overall framework is the same as this paper:
Jiahui Xu, Ling Chen, Mingqi Lv, Chaoqun Zhan, Sanjian Chen, and Jian Chang. HighAir: A
hierarchical graph neural network-based air quality forecasting method. arXiv preprint arXiv:
2101.04264, 2021.


**Summary Of The Paper:**

The paper proposes a comprehensive spatiotemporal graph model to predict air pollution concentration in the future time. The model integrates the advantages of mechanical model and machine learning. The authors compare the model with various SOTAs and the proposed method provides an impressive applicability with competitive prediction

**Summary Of The Review:**

This paper is well-written and addresses an important question possibly helping bridge environmental science and machine learning. The work falls short on the technical novelty. However, it’s still marginally above the acceptance line.

---

### Decision · Program_Chairs · 2023-01-20

**Decision:**

Reject

**Justification For Why Not Higher Score:**

A major concern is on the significance of the contribution in this paper, as most of the techniques presented such as multi-granularity graph and dynamic graph modeling are already established in other time series forecasting tasks, the technical contribution is limited.
Another concern is that the experiments are not convincing to show the effectiveness of the model. For example, the SOTA models such as informer, autoformer and N-BEATS are not presented or discussed; more ablation is needed to show that the proposed method is better than other baselines such as HighAir and PM2.5-GNN, but is currently missing in this paper.
Also,all the reviewers pointed out that the presentation of the paper needs to be improved, such as to address the missing discussion on the baseline models, the clarity of the legends on the figure, and some of the important results are only listed in the appendix (which should be moved to main paper).


**Justification For Why Not Lower Score:**

N/A

**Metareview: Summary, Strengths And Weaknesses:**

This paper studies the task of air quality prediction with spatiotemporal modeling. Specifically, the authors propose multi-granulariy spatiotemporal graph with dynamic edge modeling. Experiments are conducted on two real world datasets to show the performance gains of the proposed methods.

The idea of using a multi-granularity graph is well-motivated in this paper and the task of air quality prediction is interesting and impactful. However, as reviewers (tjqh, mQij) pointed out, the key components such as LSTM, GNN, and attention mechanisms are commonly used for time series forecasting and the multi-granularity of graphs has already been explored in the community.  More of the state-of-the-art time series forecasting models, such as informer, autoformer, N-BEATS should be included in the experiments as baselines (reviewer mQij). Also, as reviewers tjqh, VRie pointed out, the model reuses the previous framework such as HighAir but the gains compared with these previous models is not significant.

Although the authors addressed some of the reviewers’ questions, the above concerns are not sufficiently addressed. In conclusion, we suggest a rejection.


**Summary Of Ac-Reviewer Meeting:**

This is not a borderline paper.